# Retaliatory killing negatively affects African lion (Panthera leo) male coalitions in the Tarangire-Manyara Ecosystem, Tanzania

Nancy Felix[1]*, Bernard M. Kissui[2,3], Linus Munishi[1], Anna C. Treydte[1,4,5]

**1** Department of Sustainable Agriculture, Biodiversity and Ecosystem Management, The Nelson Mandela African Institution of Science and Technology, Arusha, Tanzania, **2** School for Field Studies, Center for Wildlife Management Studies, Karatu, Tanzania, **3** Tarangire Lion Project, Arusha, Tanzania, **4** Department of Physical Geography, Stockholm University, Stockholm, Sweden, **5** Ecology of Tropical Agricultural Systems, Hohenheim University, Stuttgart, Germany

\* nancyfelix25@gmail.com

**Data Availability Statement:** All data can be found under DOI: https://doi.org/10.6084/m9.figshare.20425215.v1.

## Abstract

In landscapes where people and lions coexist, conflicts are common due to livestock predation and threats to human safety. Retaliatory lion killing by humans is often a consequence and is one of the leading causes of lion population declines across Africa. We assessed the effects of retaliatory lion killing on male lion coalitions in the Tarangire-Manyara Ecosystem (TME) using a long-term dataset of lion monitoring for ten lion prides, spanning over a fourteen year-period from 2004–2018. We also interviewed 214 respondents about their attitudes and awareness of the effects of retaliatory killing on lions. We found that male lion coalitions were larger and lasted for a longer tenure period in locations with low risk of retaliatory killing, as well as far away from active hunting blocks. Further, young people (18–35 years old) had a more positive attitude towards lion existence and conservation compared to older age classes. Surprisingly, people with primary or secondary level of education were more likely to having lions killed if they attack livestock compared to people with no formal education, although the former supported lion presence for tourism in protected areas. We conclude that retaliatory killing has a large effect on long-term lion coalition dynamics and, thus, survival. Community awareness on retaliation effect varies widely, and we recommend implementing better education and policy strategies at TME to protect the declining carnivore populations.

## 1.0 Introduction

The current global decline in large carnivore populations is largely connected with an increasing human population in their geographical range leading to habitat contraction and fragmentation [1, 2]. Currently, African lion (*Panthera leo*) occupies a range of about 2.5 million km$^2$, which is only about 13% of their historical range [3]. This is lower than what was reported to be approximately 3 million km$^2$ for the year 2013 [4]. In Africa over the past 21 years, the population of African lions was around 39,373 individuals [3] but has been declining by 43%

**Funding:** The research was funded by The Rufford Foundation under Rufford Small Grant to Miss Nancy Felix with grant number 27308-1. The funds assisted in study design, data collection and analysis.

**Competing interests:** The authors have declared that no competing interests exist.

recently [3]. A recent increase in the human rural population has triggered the conversion of natural areas into agricultural fields and settlements, thus, restricting wildlife movements by blocking corridors and causing habitat fragmentation [5–7]. In most of sub-Sahara Africa, lions live both in protected areas and surrounding unprotected areas, where a decrease in their population is particularly visible [8]. In East Africa, these surrounding unprotected areas often comprises rangelands used by agro-pastoral communities which increases the interaction of local communities with lions and frequently leads to conflict [9, 10]. These conflicts are usually a result of livestock depredation, human attacks/injuries and death [11, 12]. Loss of life and property, which are linked to a reduction in family wealth, cause people to retaliate by killing lions or another carnivore [13–15]. These retaliatory killings have been shown to threaten the persistence of carnivore populations in Tanzania and Kenya, where a number of carnivores killed by man maybe proportional to the number of livestock killed by carnivores [15–18]. Yet, little is known about the extent to which retaliation affects lion populations, particularly in the long–term.

Lions are social felids, living in groups called prides that include two to eighteen related females and a coalition of one to seven males [19]. Females stay in a pride for communal rearing of offspring and males form a coalition to ensure protection, maximum reproduction, and hunting success [20]. At the age of three years and above, male lions leave their natal pride and form a coalition with brothers, cousins, or non-relatives [21]. These males roam around until they are capable of fighting males from another pride and driving them away, this is called "pride takeover" [22]. During pride takeover, a new coalition will kill any cubs less than two years old and expel sub adults of the evicted coalition to speed up female return into oestrus for reproduction [19]. Male coalitions are, therefore, important as they protect and support females in rearing their cubs to independent age [21]. Thus, the loss of any individual that forms a coalition increases the vulnerability of the entire pride and offspring [23]. The number of males in a coalition (coalition size) is an important component in ensuring the survival of the cubs, females, and the pride as a whole and can be a good indicator of population fitness [19, 20]. Further, the longer a coalition with three or more individuals can last in different prides (also known as tenure period), the more they ensure the survival of their offspring [19]. Thus, both coalition size and tenure period are a good proxy for understanding population fitness, and these two factors might strongly be compromised through retaliatory killing. Up to now, few studies have investigated how the killing of male lions affect their social structure, particularly that of male coalitions. Therefore, we quantified the impact of retaliation incidences of male lions on coalition size and tenure period and related the incidences to environmental factors as well as socio-ecological aspects.

In the Tarangire-Manyara Ecosystem, lions follow migrating prey species into communal unprotected areas during the wet season, which might be threatening lion survival [5–7]. In some unprotected areas, there are hunting blocks, areas delineated by wildlife directors for trophy hunting [24]. Hunting activities that are not properly managed, have been reported to pose adverse impacts on lion prides, particularly male coalitions [23, 25, 26]. Some male coalitions live close to these hunting block areas, which might increase their risk of being hunted. Therefore, we expected coalitions close to hunting blocks to be smaller in size and to last for a shorter tenure period than coalitions further away. Further, villages differ how they conduct retaliatory lion killing, thus we termed areas with few records of lion killings as low risk areas and vice versa, and hypothesized that male coalitions closer to high risk areas will have fewer individuals and shorter tenure period than those in the low risk village areas.

## 1.1 Community attitude on the effect of retaliation to lion population

Positive attitudes of people towards carnivore conservation depends on the benefits such as revenue, and employment opportunity [27, 28] while negative attitudes are often associated with livestock loss and human injuries/death [29–32]. In addition, socio-economic factors i.e., demography, age and gender [12, 33] as well as education and social factors [29, 34] have been reported to influence people's attitudes towards carnivores. We wanted to understand the communities' attitudes and awareness on the impact of retaliatory killing to lion population. We expected that people living in villages with high levels of livestock predation and with low level of education would be least tolerant towards livestock losses by lions and would be most likely to support retaliatory lion killing. We further hypothesized that pure pastoralists perceived lions as a greater threat than agro-pastoralist, employees, and business owners, as the former are strongly dependent on livestock only [35, 36]. Moreover, we expected people with low income to be more likely to retaliate than those with high income as was in shown [18, 37]. Also, we hypothesized that young men i.e., Maasai warriors had a higher probability of conducting retaliation compared to older men and women due to traditional activities in the Tanzanian Maasai pastoralist culture [15, 16]. Hence, our study aimed at understanding spatio-temporal dynamics of male lion coalitions in TME. Secondly, we strived to identify the effect of retaliatory killing on male lions coalitions particularly on coalition size and tenure period, found in the protected areas and surrounding unprotected areas. To achieve these objectives, we used a long-term lion monitoring dataset collected in TME from 2004 to 2018, from which male coalition size and tenure period as well as retaliatory killing frequencies were extracted. Our third objective was to assess the community attitudes towards the effect of retaliatory lion killing on the TME lion population, based on interviews in selected villages surrounding Tarangire National Park. Our paper will provide information of the effect of retaliatory killing on male lion coalitions and identify conflict hotspot areas and communication shortcomings for promoting sustainable conservation of the lion population in the TME.

## 2.0 Materials and methods

### 2.1 Study area

The study was conducted in the Tarangire-Manyara ecosystem (TME; Fig 1) with an estimated area of 35,000 km$^2$ in Northern Tanzania, covering two National Parks, several game reserves, game-controlled areas and villages [38]. About 10.0% of the TME is covered by Tarangire (2,800 km$^2$) and Manyara (330 km$^2$) National Parks (TNP and LMNP, respectively), which have an average elevation of 950–1500 m.a.s.l, average annual temperature and rainfall of 25˚C and 650 mm, respectively [6]. About 575,371 people live in Monduli (consists of the following villages Oltukai, Mswakini juu, Mswakini chini, Esilalei and Babati districts (consists of Olasiti, Minjingu, Kakoi) mostly Maasai communities involved in pastoralism and subsistence agriculture, with a 4.7% and 4.5% population growth rate, respectively [39]. This increase in human population has led to expansion of agricultural fields and livestock numbers [7].

### 2.2 Ethics statement

The research was approved by the Tanzania Wildlife Research Institute (TAWIRI) and COST-ECH with permit number 2019-345-NA-2018-354. The study was introduced in meetings to the local communities ahead of field work. A verbal consent was sought to a person before conducting an interview, as advised by the translator and most of them are not competent in writing. To ensure anonymity no names were collected and respondents' identity were coded numerically.

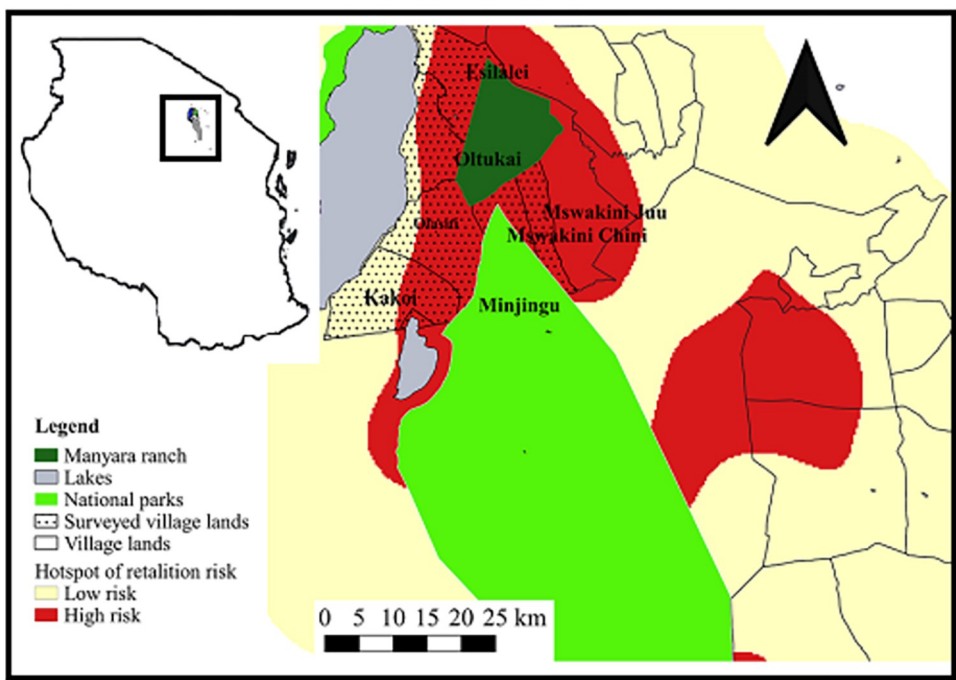

**Fig 1. Location of the surveyed village lands, surrounding protected areas, heat map of high and low retaliation risk areas in TME.** Protected areas are Manyara ranch- dark green color, National parks- light green color, lakes- grey color, surveyed villages lands- dotted black, high retaliation risk areas- red color, low retaliation risk areas- yellow color. Protected areas source shape files: Tarangire National Park Database, Villages surrounding protected areas source shape files: Tarangire Lion Project. 'Republished from [ref] under a CC BY license, with permission from [Tarangire National Park] copyright [2018]'.

## 2.3 Mapping male coalitions

We used datasets from long-term lion monitoring study by the Tarangire Lion Project (TLP) from the year 2004 to 2018. The lion population has been studied continuously since 2004 in 35,000 km$^2$ of the TME [40–42]. The number of lions in our core study area had been fluctuating around 160–200 individuals (unpublished data). We analyzed movement data of ten prides, which had been monitored for 14 years, from 2004 to 2018, and mapped pride locations using coalition data sighted at least five times within each pride. For each pride, one to two individuals (in most cases females) were fitted with a VHF-collar to track and locate lions for direct field observations [43]. For each sighted coalition, GPS coordinates were recorded, and individual males were identified. A total of 553 lion sightings for all ten prides were used in mapping pride locations, overlaying male coalition locations with village surrounding the protected areas using QGIS version 3.13.

## 2.4 Retaliation effect on male coalition

To examine the effect of retaliation on male coalitions we used extracted data of 46 male coalitions from the year 2004 to 2018. From the dataset, we recorded the number of males in a coalition and their tenure period. Each coalition was observed from when it was first sighted as resident within the pride until it was evicted by other males, i.e., the tenure period. We considered location of male coalitions i.e., whether they were inside the protected areas ("core") had migrated into surrounding unprotected areas ("periphery"). Moreover, some areas were hunting block for trophy hunting, and thus, we categorized some coalitions to be "in" or "adjacent

to" active hunting blocks. Therefore, we used the following environmental variables in our model to predict the male coalition size and tenure period for male coalition found (i) in core or periphery of the protected areas and, (ii) in high retaliation risk or low retaliation risk village land, as well as (iii) in or adjacent to hunting blocks.

## 2.5 Community perceptions on the effect of retaliation to lion population

To assess community attitude on the effect of retaliation to lion population, we surveyed seven villages in the Tarangire-Manyara Ecosystem, from March 2019 to May 2019. Villages were selected based on the frequency of retaliation occurrence (S1 Table). We used semi-structured questionnaires on a total of 214 households. In selecting an interviewee, we used systematic sampling, in which every sixth household head in each sub-village was interviewed [18]. Criteria of inclusion in the interview were; a) the household head, usually a man, had lived in the village for more than five years and b) was an adult of $\geq$ 18 years. Additionally, key informant interviews were conducted to fifteen rangers from Tarangire National Park, Burunge Wildlife Management Area, and Manyara Ranch. Information from key informants was summarized and summed up for comparison with responses from village communities. Interviews and questionnaires aimed at determining peoples' perception towards wildlife challenges and retaliatory killing incidences that occurred over the last five years. The questionnaire was tested in a pilot survey to ten individuals living in TME in February, 2019. The questionnaire was in English and was translated in Swahili or Maa by a translator. We investigated people's knowledge about the status of the current lion population, recent lion attacks and retaliatory killing events. The questionnaire had four parts; the first assessed socio-demographic information of the respondent (age, occupation, sex, education, resident time and benefits from conservation), the second part consisted of respondent awareness on wildlife related challenges and ranking of the problem animals, the third part assessed awareness, knowledge and effectiveness of measures used to protect livestock, and the last part assessed the respondent's attitude and perception towards lion populations, their trends, effect of retaliatory killing on the TME lion population and reports of lion killings in the past three years (S2 Table). The variation of community attitude towards the effect of retaliatory killing to lion population was analyzed and tested using generalized linear mixed model (GLMM) with seven fixed explanatory variables (Table 1).

## 2.6 Statistical analyses

Analyses were performed in R version 3.6.3 [44]. We calculated the mean ($\pm$SD) of male coalition size and tenure period as response variables. Data of the response variables were not normally distributed and, hence we used generalized linear mixed model (GLMM). We constructed a priori candidate models and tested the effect of retaliation on coalition size and tenure period. We used GLMM with a Poisson distribution to determine whether predictor variables of retaliation risk, location in PA, and closeness to a hunting block significantly influenced response variables. We considered interaction of the predictors retaliation risk, location in PA, and hunting block on number of males in a coalition and tenure period respectively. We created seven priori candidate models for each response variables in (S3 and S4 Tables) and regarded models with the lowest AICc and highest Akaike weights ($\omega_i$) values as the best approximating model in the set of candidate models [45]. Models with $\Delta$AICc <2 had strong support and represented a confidence set of the best model, while $\Delta$AICc values of >2 showed weak support [45].

To investigate community attitude on the effect of retaliation to lion population, we tested predictor variables age class, education, occupation, sex, resident time and benefit from

**Table 1. Identified variables for determining the effect of retaliatory killing on male coalition and lion population in TME from environmental, behavioral and socio-economic variables respectively in the year 2019.**

|  | Variable | Explanation | Variable type |
|---|---|---|---|
| **Environmental** | PA location | Location in the core part or periphery of the protected area | Categorical: in, out |
|  | Retaliation risk | Location close to (< 5 km) or far from (> 5 km) villages that had a high record of retaliation | Categorical: in, out |
|  | Hunting location | Location within active hunting blocks or not | Categorical: in, out |
| **Behavioral** | Male numbers | Number of males in a coalition group | Continuous |
|  | Tenure period | Duration a coalition group is resident in a pride | Continuous (months) |
| **Socio-economic** | Occupation | Agro-pastoralist, pure pastoralist, farmer, business owner, employee | Categorical |
|  | Age class | 18–35 yrs, 36–45 yrs, 46–55 yrs, >55 yrs | Categorical |
|  | Sex | Male, female | Categorical |
|  | Education | Illiterate, primary, secondary, tertiary | Categorical |
|  | Benefit from conservation | Employment, business opportunity, community development, no benefit | Categorical |

PA = Protected area

conservation for collinearity using corrplot package and usdm package in R (version 3.6.3) (S1 Fig). From each question, we extracted response variables that were used to construct a priori candidate models (S5 and S6 Tables). Data of the response variables were not normally distributed and, hence, we used GLMM with a Poisson distribution to analyze the variation of community attitude towards the effect of retaliatory killing on the lion population. All statistical tests were two-tailed with a 5% level of significance.

## 3.0 Results

### 3.1 Male coalitions in TME

We identified a total of 113 individual males across 10 prides (Fig 2) that belonged to 46 different coalitions and had their life history recorded until death (S7 Table). These numbers excluded the nomadic males that were sighted only once with the prides. On average (±SD), 2.4 (±1.1) (S8 Table) males older than 4 years of age formed coalitions in prides. During the study period, the largest coalition group had 5 males.

### 3.2 Effect of retaliation on coalition size

We found that male coalitions in areas with high retaliation risk had smaller group sizes with few individuals. In (S3 Table) the ΔAICc revealed that models 2 and 3 had values <2, thus constituting the confidence set of the best model. The Akaike weights ($\omega i$) showed that the best model was only slightly (1.1 times) as likely as model 2 and 3. These two models showed that if the coalition was in the periphery of the protected area and within a hunting block as well as in a village with high retaliation frequencies the coalition size was small, while male coalitions inside the PA core part were larger.

### 3.3 Effect of coalition size, lion hunting and retaliation on tenure period

The average (±SD) tenure of a coalitions within a pride was (19.9 ± 15.4) months (S2 Text) and was strongly related to the number of males and the location of the coalition. Larger coalitions located far away from areas with high risk of retaliation had a longer tenure period than those close to high retaliation risk areas. In GLMM, ΔAICc revealed that models 2, 3 and 4 had values <2 (S4 Table), thus constituting the confidence set of the best model. The Akaike weights ($\omega i$) showed that the best model was only 2.2 times as likely as models 2, 3 and 4

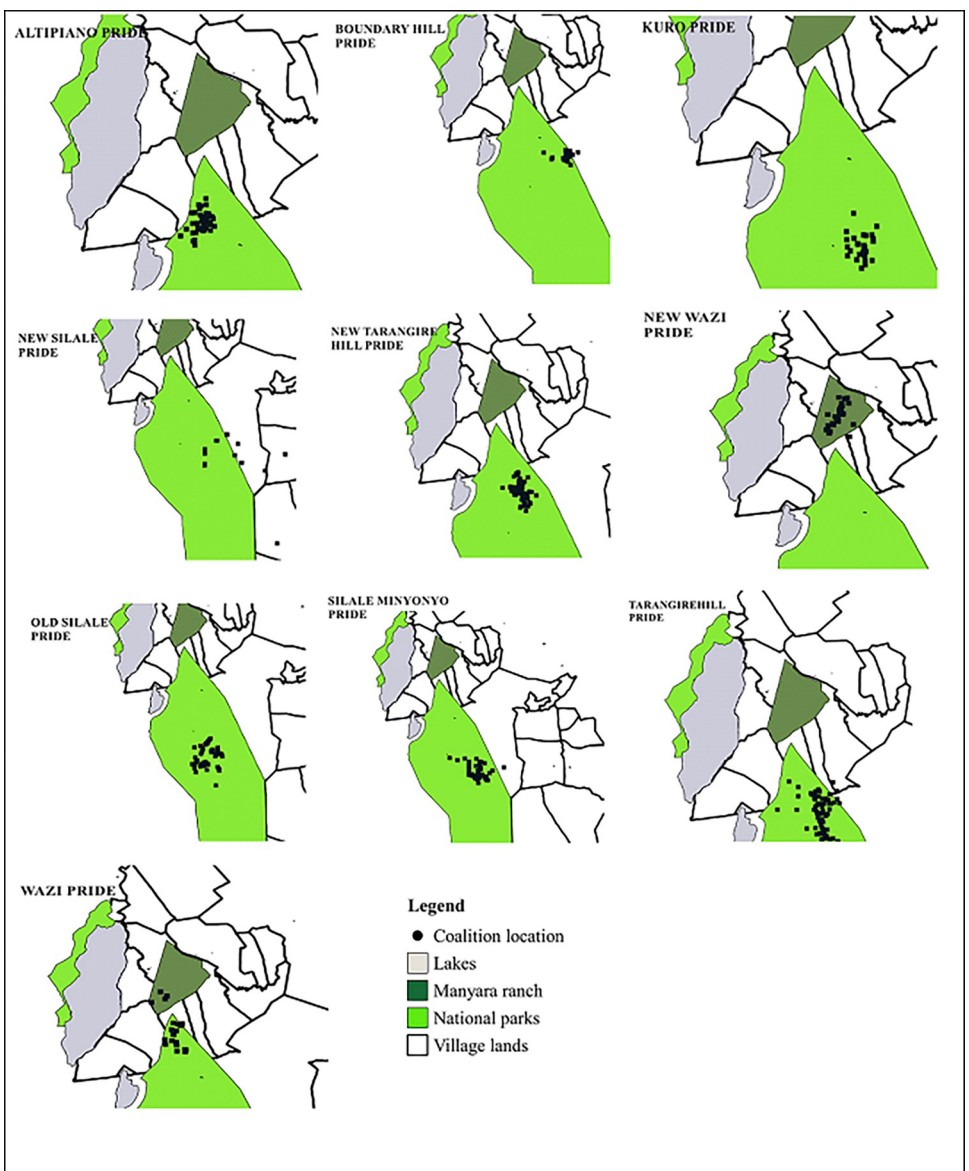

**Fig 2. Coalition locations in the protected areas and surrounding villages.** Protected areas are Manyara ranch- dark green color, National parks- light green color, lakes- grey color, village lands-plain box, coalitions location- black dots. Coalition coordinates for each pride from 2004 to 2018. Protected areas source shape files- Tarangire National park Database Shape files, Coalition coordinates source- Tarangire Lion Project. 'Republished from [ref] under a CC BY license, with permission from [Tarangire Lion Project] copyright [2018]'.

(S4 Table). From model 5 interaction of the variables revealed no effect on tenure period. Thus, number of males, PA location and retaliation influence the time period a coalition last in a pride (S8 Table).

## 3.4 Attitude of the community on the effect of retaliatory killing to lion population

A total of 214 respondents were interviewed, 55% female (n = 118) and 45% (n = 96). The respondents age range from 18 to 73 years old, with a mean of 40. Young respondents (54%)

had primary and secondary education, 2% had tertiary education, while the remaining 44% were illiterate. The average resident time for respondents at TME was 16 years. The majority of the respondents 89% (n = 191) were agro-pastoralist, 5% (n = 11) were employees, farmers 3% (n = 06), business men/women 2% (n = 04) and pure pastoralist 1% (n = 2). Half of all respondents 50% (n = 106) mentioned that they had benefited from conservation through community development that involved building of schools, hospitals and water infrastructure, while 43% (n = 91), felt that protected areas close to their villages have not been of advantage to the community. Only 5% (n = 10) had been employed as local tour guides and 3% (n = 7) have benefited through business opportunities selling ornaments to tourist.

**3.4.1 Attitude towards lion presence.**   Almost three quarters (72%) of all (n = 214) respondents from villages of both high and low retaliation risk had a positive attitude towards lion existence, while 28% had a negative attitude and did not wish lions to exist. Out of the former, 116 acknowledged that wildlife (lions) has led to natural resource protection and increased cultural tourism activities within their villages. The remaining 38 respondents had a positive attitude but suggested that wildlife should be restricted to protected areas and not allowed to migrate into villages. Our GLMM revealed that age class and education influenced the attitude of people towards lion presence (S5 Table) with younger respondents of 18–35 years old and 36- 45years, which were almost half of the interviewee 49% (n = 116) having a positive attitude towards lions (Estimate = 0.87, SE = 0.43, Z = 2.02, p = 0.04). Also, people that had no formal education were more positive towards lions than those with primary or secondary education (illiterate: Estimate = 0.34, SE = 0.32, Z = 1.06, p = <0.001; primary: Estimate = 0.03, SE = 0.34, Z = 0.11, p = 0.909; secondary: Estimate = 1.44, SE = 0.68, Z = 2.1, p = 0.035).

**3.4.2 Lion attacks on livestock and human and lion killing.**   Respondents mentioned that six lion attacks on seven people had occurred from July 2016 to May 2019, all of which happened in bushy areas and close to the protected area border. Moreover, 20 attacks on 64 individual livestock (51 cattle, 8 sheep, and 5 goats), had been reported within the last 3 years by interviewees. Of all respondents, 45% (n = 96) claimed that livestock depredation was the second most common cause of livestock loss after diseases, followed by drought. More than half (55%, n = 118) of the respondents suggested that lions and other carnivores should be killed when they attack livestock or humans, and 50% (n = 117) suggested that not only lions but any wildlife that causes damage should be killed, while 45% claimed that lions should be translocated to other areas in these cases. Most community members 42% (n = 90) were highly dependent on livestock as their source of income and they claimed that the loss of livestock by a predator threatened the family status and income. Some (15%, n = 96) respondents suggested that lion attacks on livestock had decreased while those caused by hyena (*Crocuta crocuta*) had increased. The remaining 85% claimed that livestock attacks by lions have increased. Our GLMM showed that education level significantly determined whether respondents wanted lions to be killed in return for their attacking livestock (S6 Table). Respondents with primary and secondary education were less tolerant towards livestock losses by lion compared to those without any formal education (Primary: Estimate = -0.68, SE = 0.30, Z = -2.25, p = 0.024; Secondary: estimate = -0.92, SE = 0.46, Z = -1.98, p = 0.046; Illiterate: Estimate = 0.58, SE = 0.21, Z = 2.78, p = 0.005).

**3.4.3 Retaliatory lion killing.**   Our data showed that a total of 12 lions were recorded to be killed as a revenge for livestock depredation over the last three years (July 2016 to May 2019). In addition, 15 attempts to kill lions had been organized by pastoral communities but were unsuccessful because of intervention from local government leaders, conservation officers from the National Park Authorities, Wildlife Management Areas, and Non-Governmental

Organizations. About 55% of all 214 respondents acknowledged that retaliatory killing had negative effects on lion populations while 43% suggested that it had no effect.

## 4.0 Discussion

### 4.1 Effect of retaliation and lion hunting on coalition size

Our results showed that male coalitions located in the periphery of the protected area, close to villages with high retaliation risk and in active hunting blocks had small coalition sizes as seen in prides altipiano, kuro, boundary hill, new silale, new wazi, tarangire hill and wazi (S7 Table). In TME, lions migrate into communal areas, following migrating ungulates such as wildebeest (*Connochaetes taurinus*), zebra (*Equus burchelli*) during wet season [7, 15]. Hence, they spend about 6 months outside of protected areas and become vulnerable to retaliation due to livestock predation.

In accordance with [23], we found that lion hunting affected coalition sizes negatively, which reduces social stability within a pride and causes cascading effect to lion abundance at large [23, 46, 47]. This aligns with our hypothesis that male coalitions close to hunting blocks and villages with high retaliation risk would be small. With respect to male coalition responsibility in prides, small coalition groups are at risk of not ensuring protection against intruders and persistence of the pride. This increases the risk of pride take over by other intruder males in areas where male competition is high [19]. Not only local hunting might reduce lion fitness, in Zimbabwe the lion population outside Hwange National Park faced a continuous decline due to trophy hunting [48]. In northern Tanzania, trophy hunting around Serengeti and Ngorongoro Conservation Area, showed to have devastating impact on lion population [23, 26].

### 4.2 Influence of coalition size and location on tenure period

Our findings showed that tenure period of male coalitions depends on the number of males in a coalition and its location in a protected area, as predicted by our expectation. Prides in the periphery with coalition size of two males had a short tenure period less than 12 months due to retaliation and hunting when compared to prides in the midst of the PA with similar coalition size or even singleton, lasting more than 12 months likely due to absence of human disturbance. This is similar to studies conducted in northern Tanzania [19, 20], where large coalition groups had a significant advantage of having longer tenure period within prides, also ensuring pride and offspring survival. Male coalitions will remain longer in a pride of females located in areas with adequate habitat quality, prey abundance and water sources (Personal Observation). In Zambia, at Kafue National Park, both female and male lions that had their home range closer to the border of the park disappeared due to trophy hunting and retaliation [49, 50], supporting our results of reduced coalition size and tenure period at the periphery.

### 4.3 Community attitude and awareness on the effect of retaliatory killing to lion population

We found that the attitudes of the local community towards carnivores, in particular lions, were positive as long as lions stayed away from human settlements. Our finding that younger people have a more positive attitude towards lions, as do the illiterate people is contrary to what we had expected, also is in contrary to other studies as well. In TME, most of Maasai warriors have primary and secondary level of education and they are taught about wildlife conservation in schools [36]. But, according to our findings most of the young people feel that they are not involved in resolving wildlife related problems, which might have led to their negative attitude.

Engaging young people in resolving human-lion conflicts by using them as guards for wild-life presence in communal areas might stimulate a more positive attitude as evidence in the Lion Guardian and Ruaha carnivore project [51]. These activities are aimed at broadening their exposure to direct benefit from conservation through employment and will increase tolerance to losses of livestock by lions [12, 36, 52]. We also found that people, who had lived in the villages for more than ten years, learnt ways to coexist with wildlife by fortifying their livestock enclosure using chain-linked fence and adults guarding livestock in risk areas where carnivore exists [53]. Our key informants mentioned that the positive attitude by the communities towards lion conservation has increased in the last three years. Further, rangers in the Tanzania National Parks noted an increase in women groups that were engaged in cultural tourism which promote their income and positive attitudes. In northern Tanzania, particularly around Tarangire National Park people have negative perceptions towards carnivores and conservation because of livestock losses to predators while a positive perception is often associated with benefits received from tourism activities and sport hunting [14, 15]. This aligns with findings from our study where people understood the importance of wildlife and associated benefits but still wishes wildlife to be controlled and remote from humans.

In our study, respondents reported livestock attacks by lions occurred most often in the bush/ grazing fields that are close to park boundaries while, attacks by hyena (*Crocuta crocuta*) were common in the enclosures/boma which is similar to what was reported by [15, 53, 54]. From the study site, conflicts between human and lions mainly occurred during the rainy seasons and thus, can be predictable so that measures can be taken to reduce livestock loss and associated lion retaliation. This involves regularly monitoring the locations of lions, particularly when they roam outside of the parks and conducting frequent patrols in the identified conflict hotspot areas. These activities can help alerting communities on the presence of predators in their settlements in order to increase vigilance by both protected area officers and community members for protecting lions and livestock.

We found that reported lion retaliation incidences had declined during the study period by 2019. In recent years, more attempts of retaliation had been stopped through the cooperation of government officials and non-government organizations, who received information from local government leaders and informers living in the village areas (pers. comm). This agrees with Mkonyi et al., 2017 [53] that fewer people engage in retaliation, likely due to fewer cases of livestock attacks by lions. Moreover, an increase in recent studies about lions in TME has broadened the knowledge of the study population [53, 55–57].

## Conclusion

Our study has identified hotspot areas where lion retaliation has occurred over the past fourteen years and set these hotspots in relation to environmental and socio-economic factors. We found that retaliatory killing negatively impacts number and tenure of male lion coalitions, which are crucial for protecting the entire lion pride and ensuring survival of the cubs. We further found that small male coalitions have shorter tenure periods than larger coalitions. Further, we conclude that the location of a particular male coalition influences its tenure period. We highlight that retaliatory killing negatively affects lion social structure as well as long-term lion survival and that awareness-raising and strategies (i.e., policies) need to be established and implemented at TME to protect the declining lion population as pointed out by Mkonyi et al., 2017 [53]. The local community around TME had little knowledge on the effect of retaliatory killings on the lion populations, highlighting the need for better communication and awareness raising among local communities, conservation agencies and park management. In our study, many local communities acknowledged that the number of livestock attacks by lions

have decreased over the last years, compared to other carnivore species such as hyena (*Crocuta crocuta*). Lastly, the attitude of people towards lions and other carnivores was dependent on the level of education and age, therefore, we suggest more environmental education programs, particularly around protected areas.

## Supporting information

**S1 Fig. Matrix of Spearman rank correlation coefficient for predictor variables.**
(DOCX)

**S1 Table. Villages found in TME with a record of lion retaliation from 2004 to 2018.**
(DOCX)

**S2 Table. Questionnaire in English and Kiswahili to key informants and households.**
(DOCX)

**S3 Table. Effect of retaliation on coalition size.**
(DOCX)

**S4 Table. Effect of retaliation on tenure period.**
(DOCX)

**S5 Table. Attitude of the community whether lions have right to live or not.**
(DOCX)

**S6 Table. Attitude of the community whether lions should be killed after livestock depredation.**
(DOCX)

**S7 Table. Summary of male lion coalition groups in TME.**
(DOCX)

**S8 Table. Summary of the parameter estimate for tenure period.**
(DOCX)

**S1 Text. Summary of basic statistics of males number.**
(XLSX)

**S2 Text. Summary of the basic statistic of the tenure period.**
(XLSX)

**S3 Text. Data for the questionnaire survey of the communities around TME.**
(XLSX)

**S4 Text. Data of male coalitions identified in TME.**
(XLSX)

## Acknowledgments

We thank the long-term studies by the Tarangire Lion Project which was supported by African Wildlife Foundation, Panthera Foundation, Peoples Trust for Endangered Species, WildAid. We are grateful for provision of research permits from COSTECH, TAWIRI and Regional Officers for particular district. And we sincerely appreciate the respondents' cooperation during the survey.

## Author Contributions

**Conceptualization:** Nancy Felix, Bernard M. Kissui, Linus Munishi, Anna C. Treydte.

**Data curation:** Nancy Felix.

**Formal analysis:** Nancy Felix, Bernard M. Kissui, Linus Munishi, Anna C. Treydte.

**Funding acquisition:** Nancy Felix, Bernard M. Kissui.

**Investigation:** Nancy Felix.

**Methodology:** Nancy Felix, Bernard M. Kissui, Linus Munishi, Anna C. Treydte.

**Project administration:** Nancy Felix, Bernard M. Kissui, Linus Munishi, Anna C. Treydte.

**Resources:** Nancy Felix, Bernard M. Kissui.

**Software:** Nancy Felix.

**Supervision:** Bernard M. Kissui, Linus Munishi, Anna C. Treydte.

**Validation:** Nancy Felix, Bernard M. Kissui, Linus Munishi, Anna C. Treydte.

**Visualization:** Nancy Felix, Bernard M. Kissui, Linus Munishi, Anna C. Treydte.

**Writing – original draft:** Nancy Felix, Bernard M. Kissui, Linus Munishi, Anna C. Treydte.

**Writing – review & editing:** Nancy Felix, Bernard M. Kissui, Linus Munishi, Anna C. Treydte.

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
