## [Decision Letter · Decision Letter 0]

9 Nov 2021

PONE-D-21-29635Retaliatory killing negatively affects African lion (Panthera leo) male coalitions in the Tarangire-Manyara Ecosystem, Tanzania.PLOS ONE

Dear Dr. Felix,

Thank you for submitting your manuscript to PLOS ONE. After careful consideration, we feel that it has merit but does not fully meet PLOS ONE’s publication criteria as it currently stands. Therefore, we invite you to submit a revised version of the manuscript that addresses the points raised during the review process.

Your submission provides useful information that would be welcome in the literature. However, the text needs significant re-writing before it can be considered for publication. Below you will find reviews from two reviewers, as well as a review from me. Please note that revision does not guarantee publication.

We look forward to receiving your revised manuscript.

Kind regards,

Stephanie S. Romanach, Ph.D.

Academic Editor

PLOS ONE

Journal Requirements:

2. Please include additional information regarding the survey or questionnaire used in the study and ensure that you have provided sufficient details that others could replicate the analyses. For instance, if you developed a questionnaire as part of this study and it is not under a copyright more restrictive than CC-BY, please include a copy, in both the original language and English, as Supporting Information. If the original language is written in non-Latin characters, for example Amharic, Chinese, or Korean, please use a file format that ensures these characters are visible

3.We note that Figures 1 in your submission contain map images which may be copyrighted. All PLOS content is published under the Creative Commons Attribution License (CC BY 4.0), which means that the manuscript, images, and Supporting Information files will be freely available online, and any third party is permitted to access, download, copy, distribute, and use these materials in any way, even commercially, with proper attribution. For these reasons, we cannot publish previously copyrighted maps or satellite images created using proprietary data, such as Google software (Google Maps, Street View, and Earth). For more information, see our copyright guidelines: http://journals.plos.org/plosone/s/licenses-and-copyright.

1. You may seek permission from the original copyright holder of Figures 1 to publish the content specifically under the CC BY 4.0 license.  

Additional Editor Comments:

Overall, this manuscript needs to be reviewed for English language fluency. Throughout the text, some of the phrases and sentences are unclear. I point out several of those areas below, but there are many more. Importantly, you are missing a lot of information for your reader to be able to follow what you did in your study and what resulted from it. Additionally, superfluous information is added in places making the manuscript more confusing to follow. It might be worth looking at other published manuscripts to understand the clarity of detail that needs to be provided for a submission to PLOS ONE.In the Introduction, the structure is somewhat jumbled. It needs to be re-organized to flow from the general to the more specific, or at least sticking with a clear topic within a paragraph. Currently the text jumps around, and includes information that should be removed, reorganized, or moved to other sections.Some of your citations in the text are misleading in that the authors you cite are not specifically writing about the topics you cite them for. Take care to cite properly.In your Results, you provide outputs from your models but not the basic statistics to allow your readers to understand the conclusions you discuss later in the paper. This is a critical publication criterion for PLOS ONE -- your reader needs to be able to follow you through to your conclusions, which currently is not the case. For example, you discuss negative impacts on coalitions, but you do not provide the results to support this.Further, you do not link your discussion back to all of your model results. If you are going to keep those analyses, use them in your text. If not, remove them. Depending on which results you want to focus on, you might want to reconsider which results go in Supplemental and which belong in the main text.Your Discussion provides interesting information but does not focus on discussing your results as much as bringing in evidence from other studies. For example, in your Conclusion, you state that retaliatory killing has negative impacts on coalitions but you don’t discuss this idea. It’s good that you are comparing your work with other studies, but you need to first focus on your own results before the reader can follow you in comparing to others’ results.In terms of clarity, some of your uses of “/” or “i.e.” within a sentence should be removed. In these instances, think through what message you are trying to convey and write in plain words.As a side note, it might be worth being aware of this paper,Romanach, S.S., Lindsey, P.A. and Woodroffe, R., 2007. Determinants of attitudes towards predators in central Kenya and suggestions for increasing tolerance in livestock dominated landscapes. Oryx, 41(2), pp.185-195.-
Line 55: consider changing “open” to “natural”-
L 59: change to, “in most of”-
L 68: social grouping of what? Use caution when you write “nothing has been studied” unless you are certain this is the case. Your phrasing starting at the end of L 86 is preferable.-
L 75: you state above that prides are females so be clear in the text what the role of males is, and perhaps clarify the relationship between prides and coalitions in L 70-
L 90: you need to define what makes an area high vs low risk-
L 94: adjust to give the name then the citation number after-
L 93-98: tighten up and clarify this section. The idea of hunting comes mixed in with retaliatory killings and will not be clear to the reader that you have now switched to trophy hunting at first mention of hunting.-
Additionally, in the same lines above, what you included in your analyses should go in the Methods, not your Introduction-
L 99: tell your reader what a hunting block is, many will not be familiar with trophy hunting lingo-
L 113: where were the data collected? TME?-
L 114: it’s not clear what you mean by “coalition groups” in this context. You can give the details in the Methods, but as a general term, it’s not clear if you mean size of the group other metrics-
L 118: what specifically about the social group? Group size?-
L 181: clarify how key informant interview data were used with respect to your analyses. How many of these interviews were conducted? How were the results used?-
L 358-360: say how these factors are influenced by location, or remove and start with your second sentence-
L 362: be clear that this is trophy hunting. Readers might not distinguish between retaliatory killing and sport hunting as both involve hunting in some form. Be clear in the Introduction and then use consistent terminology throughout.-
Starting L 364: it’s not clear why you are bringing in other causes of population decline (e.g., snaring). Be clear in how you are connecting these studies to yours.-
L 365: state the country locations for these two parks-
L 373: This is not clear. Who did the attacking? And on what? Sub-adult male lions were killed so they did not try to take over prides as they got older? This connection is not clear.-
L 374: Again, say who did this and then give the citation-
L 376: please be more specific in what you mean by “human habitats” or use another term-
L 377: whose stability?-
L 387: clarify that you are comparing your results to a study in another country-
L 389: perhaps use “supporting” instead of “conforming”-
L 391-392: adjust section headings for proper grammar, here and throughout-
L 408: “humans”-
L 410 - 441: this is the Discussion, be sure you are discussing (here and elsewhere) what your findings mean, not just stating or re-stating them. Same with the Conclusion. You make general statements about negative impacts, but you don’t discuss what these are or what they mean.-
L 431: what determines risk? This needs to be clear from the beginning for your readers to understand your findings. Is this related to the information in SI Table 1?-
L 439: which other carnivores? You brought up the idea of “carnivores” previously in the text, but I do not see species other than lions being investigated in your study-
Figure 1. Is very cluttered and difficult to read. The background colors make it challenging to read the labels. Areas you mention (L 131) are not labeled but others are. Consider what is necessary to label (e.g., location names, pride names given in the text) and what text can be removed. The image is of poor quality to read on enlargement. Not all colors are labeled in the legend. It might be clearer to focus on the prides in Fig 2 as you do and provide the other basic information in Fig 1. You could try substituting the points for polygons for the pride ranges to see if it makes the figure less cluttered.Table 5: final column header not displaying properly-
Table 7: fix formatting-
Citation 27, my last name truncated. It should read as Romanach.

Reviewers' comments:

Reviewer's Responses to Questions

**Comments to the Author**

1. Is the manuscript technically sound, and do the data support the conclusions?

Reviewer #1: Yes

Reviewer #2: Partly

2. Has the statistical analysis been performed appropriately and rigorously? 

Reviewer #1: Yes

Reviewer #2: Yes

3. Have the authors made all data underlying the findings in their manuscript fully available?

Reviewer #1: Yes

Reviewer #2: No

4. Is the manuscript presented in an intelligible fashion and written in standard English?

Reviewer #1: Yes

Reviewer #2: Yes

5. Review Comments to the Author

Reviewer #1: I did not find any mention on the gender aspect regarding human - lion conflicts

Reviewer #2: This is an interesting paper addressing an important, yet understudied topic. My "no" response to question 3 is in reference to not seeing the full set of underlying data, from lion population as well as community responses. Further, it seems that much of the analysis on community perception focuses only on a set of the questionnaire data shown in Table 2. Specifically, questions 17-22 seem to provide rich qualitative data but are not addressed in the analysis, results, or discussion sections of the paper. Were the findings from these questions not significant? Or were they incorporated but not described well enough?

There are only two mentions of the factor of human communities blocking migration corridors, leading to negative impacts on the lion populations - Did your QGIS analysis consider an increase in fragmentation and thus an increase in edge habitat, which seems to be the locations where conflict occurs most?

The description of community demographics do not speak to the sex factor, and I do not see a breakdown of your "predictors" in Table 2. Specifically, because you limited your interviews to heads of household, and not knowing enough about the gender norms of leadership in these communities, I wonder if the study group was biased towards males or towards females, and if you gathered insight into how sex might influence lion perception?

I also do not see from the information provided how "benefit" was decided - did a person specify whether they fell into one of the factors within this category?

Besides these questions regarding the inclusion of all available data on community perception, there are a few editorial comments mainly in the conclusion section. The sentence that begins on line 394 is not complete. Please check that other grammatical edits are completed.

6. PLOS authors have the option to publish the peer review history of their article (what does this mean?). If published, this will include your full peer review and any attached files.

Reviewer #1: **Yes: **Cuthbert Leonard Nahonyo

Reviewer #2: No

---

## [Author Response · Author response to Decision Letter 0]

7 Feb 2022

Thank you Editors and Reviewers for the comments

---

## [Editor Report · Decision Letter 1]

14 Feb 2022

PONE-D-21-29635R1Retaliatory killing negatively affects African lion (Panthera leo) male coalitions in the Tarangire-Manyara Ecosystem, Tanzania.PLOS ONE

Dear Dr. Felix,

Thank you for submitting your manuscript to PLOS ONE. However, you have not submitted the Revised Manuscript with Track Changes. Please include this with your re-submission.

We look forward to receiving your revised manuscript.

Kind regards,

Stephanie S. Romanach, Ph.D.

Academic Editor

PLOS ONE
---

## [Author Response · Author response to Decision Letter 1]

1 May 2022

I have included the rebuttal letter that responds to each question raised by reviewers , also a marked up copy of the revised manuscript with track changes and un-marked copy of the manuscript.

---

## [Editor Report · Decision Letter 2]

18 Jul 2022

Retaliatory killing negatively affects African lion (Panthera leo) male coalitions in the Tarangire-Manyara Ecosystem, Tanzania.

PONE-D-21-29635R2

Dear Dr. Felix,

We’re pleased to inform you that your manuscript has been judged scientifically suitable for publication and will be formally accepted for publication once it meets all outstanding technical requirements.

Kind regards,

Stephanie S. Romanach, Ph.D.

Academic Editor

PLOS ONE
---

## [Editor Report · Acceptance letter]

4 Aug 2022

PONE-D-21-29635R2 

Retaliatory killing negatively affects African lion (Panthera leo) male coalitions in the Tarangire-Manyara Ecosystem, Tanzania 

Dear Dr. Felix:

I'm pleased to inform you that your manuscript has been deemed suitable for publication in PLOS ONE. Congratulations! Your manuscript is now with our production department. 

Kind regards, 

on behalf of

Dr. Stephanie S. Romanach 

Academic Editor

PLOS ONE